# BKPyV DNAemia in Kidney Transplant Recipients Undergoing Regular Screening: A Single-Centre Cohort Study

**DOI:** 10.3390/microorganisms12010065

**Published:** 2023-12-29

**Authors:** Daniel B. Rasmussen, Dina L. Møller, Sebastian R. Hamm, Álvaro H. Borges, Alex C. Y. Nielsen, Nikolai S. Kirkby, Søren S. Sørensen, Susanne D. Nielsen

**Affiliations:** 1Viro-Immunology Research Unit, Department of Infectious Diseases 8632, Rigshospitalet, University of Copenhagen, 2100 Copenhagen, Denmark; daniel.braeuner.rasmussen@regionh.dk (D.B.R.); dina.leth.moeller@regionh.dk (D.L.M.); sebastian.rask.hamm.02@regionh.dk (S.R.H.); 2Department of Infectious Disease Immunology, Statens Serum Institut, 2300 Copenhagen, Denmark; albs@ssi.dk; 3Centre of Excellence for Health, Immunity and Infections (CHIP), Rigshospitalet, University of Copenhagen, 2100 Copenhagen, Denmark; 4Department of Clinical Microbiology, Rigshospitalet, University of Copenhagen, 2100 Copenhagen, Denmark; alex.christian.yde.nielsen@regionh.dk (A.C.Y.N.); nikolai.kirkby@regionh.dk (N.S.K.); 5Department of Nephrology, Rigshospitalet, University of Copenhagen, 2100 Copenhagen, Denmark; soeren.schwartz.soerensen@regionh.dk; 6Department of Clinical Medicine, Rigshospitalet, University of Copenhagen, 2100 Copenhagen, Denmark

**Keywords:** BKPyV DNAemia, BK viremia, kidney transplant, cohort study, opportunistic infection, immunosuppression, graft function, graft loss

## Abstract

Infection with BK polyomavirus (BKPyV) is a common opportunistic infection after kidney transplantation (KT) and may affect graft function. We aimed to determine the incidence, risk factors, and clinical outcomes of BKPyV DNAemia in a prospective cohort of 601 KT recipients transplanted from 2012 to 2020. BKPyV PCR on plasma was performed at days 60, 90, 180, 270, and 360 post-KT. Any BKPyV DNAemia was defined as a single BKPyV DNA of ≥1000 copies/mL. Severe BKPyV DNAemia was defined as two consecutive BKPyV DNA of ≥10,000 copies/mL. Cumulative incidences were investigated using the Aalen–Johansen estimator, and the risk factors were investigated in Cox proportional hazard models. The incidence of any BKPyV DNAemia and severe BKPyV DNAemia was 21% (18–25) and 13% (10–16) at one year post-KT, respectively. Recipient age > 50 years (aHR, 1.72; 95% CI 1.00–2.94; *p* = 0.049), male sex (aHR, 1.96; 95% CI 1.17–3.29; *p* = 0.011), living donors (aHR, 1.65; 95% CI 1.03–2.74; *p* = 0.045), and >3 HLA-ABDR mismatches (aHR, 1.72; 95% CI 1.01–2.94; *p* = 0.046) increased the risk of severe BKPyV DNAemia. Any BKPyV DNAemia was associated with an increased risk of graft function decline (aHR, 2.26; 95% CI 1.00–5.12; *p* = 0.049), and severe BKPyV DNAemia was associated with an increased risk of graft loss (aHR, 3.18; 95% CI 1.06–9.58; *p* = 0.039). These findings highlight the importance of BKPyV monitoring post-KT.

## 1. Introduction

Immunosuppressive therapy is required after kidney transplantation (KT) to prevent graft rejection. However, the use of immunosuppressive agents in KT recipients increases the risk of viral infections, such as BK polyomavirus (BKPyV) [1,2]. BKPyV can cause BKPyV DNAemia, which has been associated with graft impairment [3,4,5,6,7].

BKPyV DNAemia occurs in 10–30% of KT recipients within the first year post-KT [8,9,10,11,12,13,14], with heavily immunosuppressed recipients being at greater risk [12,15]. BKPyV DNAemia can progress to BKPyV-associated nephropathy (BKPyVAN) in 5–10% of cases [16], which has been linked to impaired graft function and an increased risk of graft failure [3,4,5,6,7]. There is no antiviral treatment for BKPyV, and management is mainly reduction of the immunosuppressive treatment [17]. Screening for BKPyV DNAemia to facilitate early detection is recommended by the International Society of Nephrology [18]. When BKPyV DNAemia exceeds 10,000 copies/mL, reducing immunosuppressive therapy is recommended [18]. However, even BKPyV DNAemia below this threshold may impact graft function [19].

Previous studies have determined the prevalence and impact of BKPyV DNAemia in KT recipients [2,8,9,10,11]. However, most of these studies are limited by a small cohort size, lack of routine screening, or retrospective design. At our center, a systematic prospective screening for BKPyV DNAemia with quantitative PCR within the first year post-KT was introduced in 2012, allowing for the early detection of asymptomatic BKPyV DNAemia [20]. We aimed to determine the incidence, risk factors, and clinical outcomes of BKPyV DNAemia in a large, prospective cohort of KT recipients undergoing routine screening.

## 2. Materials and Methods

### 2.1. Study Design and Participants

In this study, we included 601 adult KT recipients (≥18 years of age) undergoing transplantation at Copenhagen University Hospital, Rigshospitalet, Denmark, from 1 January 2012 until 1 January 2020. Recipients with simultaneous pancreas–kidney or liver–kidney transplantation were excluded. Recipients were identified from The Knowledge Centre for Transplantation database, which includes all KT recipients transplanted at Rigshospitalet since 1 January 2010. KT recipients were followed from transplantation until graft loss, death, or two years post-KT, whichever came first. We included BKPyV screening results from the first year post-KT and investigated the clinical outcomes, including graft function decline, graft loss, and death within the first two years post-KT. Recipients may contribute with multiple transplantations in the case of re-KT within the inclusion period.

### 2.2. Data Sources

KT recipient characteristics and outcomes, including recipient age, sex, disease leading to KT, graft biopsy diagnoses, including Banff classification [21], and immunosuppressive treatment when discharged post-KT, were obtained from medical records. The outcomes obtained were the estimated glomerular filtration rate (eGFR) measurements, return to dialysis, graft loss, re-KT, and death.

Quantitative PCR results were retrieved from the Danish Microbiology Database (MiBa). MiBa has complete coverage of all microbiologic samples collected in general practice and hospitals in Denmark since 2010 [22].

Donor- and graft-related variables were retrieved from the ScandiaTransplant’s clinical registry containing transplant-related data from all solid organ transplantations performed in Denmark since 1994 [23]. These data included donor age, donor type, cold ischemia time, ABO incompatibility (ABOi), and human leukocyte antigen (HLA)-ABDR matching.

The retrieval of the data was approved by the National Committee on Health Research Ethics (H-170024315) and the Data Protection Agency (RH-2016-47).

### 2.3. Immunosuppression and CMV Prophylaxis

The standard induction therapy consisted of basiliximab and methylprednisolone. For patients judged as high risk for rejection, anti-thymocyte globulin was used instead of basiliximab. Induction therapy for ABOi living donor KT consisted of rituximab pre-KT and intravenous immunoglobulin together with basiliximab.

Maintenance immunosuppression usually consisted of triple therapy including tacrolimus, mycophenolate mofetil (MMF), and prednisolone. In KT recipients with a history of adverse reactions to tacrolimus, this was replaced by cyclosporine (CyA). Calcineurin inhibitors were replaced by mammalian target of rapamycin (mTOR) inhibitors in cases of hemolytic uremic syndrome.

All KT recipients received 3 months of cytomegalovirus (CMV) chemoprophylaxis post-KT. Recipients at high to intermediate risk received 450 mg Valganciclovir daily, whereas recipients at low risk received 450 mg Valganciclovir every second day, with a reduction in dosage depending on renal function.

Ureteral stents were placed during KT and removed by cystoscopy approximately two weeks post-KT.

### 2.4. BK Polyomavirus

Any BKPyV DNAemia was defined as a single BKPyV DNA of ≥1000 copies/mL measured by PCR in accordance with the guidelines provided by the American Society of Transplantation Infectious Diseases Community of Practice [24]. High BKPyV DNAemia was defined as a single BKPyV DNA of ≥10,000 copies/mL. Severe BKPyV DNAemia was defined as two consecutive BKPyV DNA of ≥10,000 copies/mL. Mild BKPyV DNAemia was defined as a single BKPyV DNA of ≥1000 copies/mL, without progression to severe BKPyV DNAemia.

Screening for BKPyV DNA in KT recipients was part of the Management of Post-transplant Infections in Collaborating Hospitals (MATCH) program [20]. MATCH was established at Rigshospitalet in 2011 to improve the management of infections post-transplantation. This screening program included BKPyV PCR tests on plasma samples collected on days 60, 90, 180, 270, and 360 post-KT [20]. BKPyV PCR tests were performed using the REALstar BKV PCR Kit 1.0 (Altona Diagnostics, Hamburg, Germany). Following the detection of BKPyV DNAemia, additional BKPyV PCR tests were conducted every two weeks until a decrease in viral load was observed.

BKPyV DNAemia was managed by a reduction in immunosuppressive therapy individualized based on the patient’s clinical condition and response, usually following the scheme (A) reduction of MMF, (B) reduction of prednisolone, (C) reduction of calcineurin inhibitors, and (D) conversion from tacrolimus to CyA.

BKPyVAN was defined as any BKPyV DNAemia in combination with a SV40-positive biopsy.

### 2.5. Definitions

Cold ischemia time was defined as the time between the chilling and removal of the donor kidney and the time of restored blood supply in the recipient. Graft rejection was defined as any biopsy-proven rejection treated with a minimum of 250 mg methylprednisolone in three days. Graft function decline was defined as either a return to persistent dialysis or a 50% decline in eGFR from day 90 post-KT until end of follow-up. Graft loss was defined as a return to persistent dialysis or graftectomy.

### 2.6. Statistics

Continuous data were presented as medians with an interquartile range (IQR) and categorical data as proportions. Differences in numeric variables were assessed by the Mann–Whitney *U* test or *t*-test, depending on the distribution of data. Differences in categorical variables were assessed by Fisher’s exact or chi-square, depending on the number of outcomes associated with each variable.

In order to account for different KT recipient follow-up times, our results were investigated through a survival analysis. Cumulative incidences were calculated using the Aalen–Johansen estimator with graft loss and death as competing risks (R-package prodlim).

Cox proportional hazards models were used to identify clinical and demographic risk factors independently associated with any BKPyV DNAemia and severe BKPyV DNAemia (R-package survival). This analysis was conducted in both a univariable and a multivariable model. To account for potential confounding, we adjusted for recipient sex, donor age, donor type, ABOi, cold ischemia time, and type of calcineurin inhibitor. To ensure unidirectional causality, rejection episodes prior to BKPyV DNAemia were included as a time-dependent variable. The risk of graft function decline, graft loss, and death following BKPyV DNAemia were investigated in Cox proportional hazards models. In this specific model, BKPyV DNAemia was treated as a time-dependent covariate, ensuring that BKPyV DNAemia preceded all observed outcomes.

Results with a *p*-value below 0.05 were considered statistically significant. All analyses were conducted using statistical software R version 3.6.1 (R Foundation for Statistical Computing, Vienna, Austria) [25].

## 3. Results

### 3.1. Patient Characteristics

We included 601 KT recipients during the study period. Prior to the first BKPyV screening, 31 recipients were censored, as five recipients died (1%) and 26 experienced graft loss (4%). The remaining 570 KT recipients were investigated with a total of 1111 person-years of follow-up. The mean recipient age was 50 years (IQR 41–61), and 64% of KT recipients were males. First-time kidney transplants constituted 87% of the transplants. Glomerulonephritis (26%) and cystic kidney disease (17%) were the most common causes of KT (Table 1).

Data on HLA-ABDR mismatches were available for 558 (98%) KT recipients. A higher degree of HLA-ABDR mismatching was observed in the KT recipients with any BKPyV DNAemia who progressed to severe BKPyV DNAemia. Furthermore, HLA-ABDR mismatches were more common in KT recipients receiving an allograft from living compared to diseased donors (median 3 (IQR 2–4) vs. 3 (IQR 2–3), *p* < 0.001).

Rejections requiring methylprednisolone (MP) treatment were observed in 143 recipients (25%) within the first year post-KT. In 19 instances, the rejections preceded BKPyV DNAemia, occurring with a median interval of 86 (57–150) days from rejection to the onset of DNAemia.

### 3.2. Screening Results

Within the first year post-KT, 3577 BKPyV PCRs were conducted on 570 KT recipients, with a median of 5 (IQR 4–7) and a minimum of one BKPyV PCRs performed per recipient (Table 2). In total, 148 recipients (26%) were tested less than the scheduled five times, and 35 recipients (6%) were tested less than three times. The screening protocols were discontinued due to six graft losses and seven deaths within the first year post-KT. The remaining were not fully tested due to unknown reasons.

In total, any BKPyV DNAemia was found in 129 KT recipients, and the cumulative incidence of any BKPyV DNAemia was 21% (95% CI 18–25) (Figure 1). Among KT recipients with any BKPyV DNAemia, 80 recipients developed severe BKPyV DNAemia, corresponding to a cumulative incidence of 13% (95% CI 10–16). The median peak viral load for all KT recipients with any BKPyV DNAemia was 1.7 × 10^5^ (IQR 2.6 × 10^4^–1.9 × 10^6^), and the viral load peaked 128 (IQR 103–195) days post-KT.

### 3.3. Risk Factors for BKPyV DNAemia

Recipient- and graft-related characteristics associated with BKPyV DNAemia were investigated in time-dependent Cox proportional hazard models (Figure 2). In the unadjusted model, an increased risk of any BKPyV DNAemia was observed in recipients with age > 50 years (HR, 1.74; 95% CI 1.21–2.51; *p* = 0.003) and donor age > 50 years (HR, 1.46; 95% CI 1.02–2.09; *p* = 0.045). In the model adjusted for recipient sex, donor age, donor type, ABOi, cold ischemia time, and maintenance immunosuppression, recipient age > 50 years (aHR, 1.75; 95% CI 1.16–2.63; *p* = 0.007) remained associated with an increased risk of any BKPyV DNAemia, whereas donor age did not increase the risk. Recipient sex, donor type, ABOi, cold ischemia time, tacrolimus-based maintenance immunosuppression, and rejections were not associated with an increased risk of any BKPyV DNAemia.

Likewise, the risk factors for severe BKPyV DNAemia were investigated (Figure 2). In the unadjusted model, recipient age > 50 years (HR, 1.78; 95% CI 1.09–2.89; *p* = 0.020), male sex (HR, 2.03; 95% CI 1.21–3.40; *p* = 0.007), and donor age > 50 years (HR, 1.77; 95% CI 1.11–2.82; *p* = 0.016) were associated with an increased risk of severe BKPyV DNAemia. In the adjusted model, the living donor type increased the risk of severe BKPyV DNAemia (aHR, 1.65; 95% CI 1.03–2.74; *p* = 0.045), and both recipient age > 50 years (aHR, 1.72; 95% CI 1.00–2.94; *p* = 0.049) and male sex (aHR, 1.96; 95% CI 1.17–3.29; *p* = 0.011) remained associated with an increased risk of severe BKPyV DNAemia. In this model, donor age, ABOi, cold ischemia time, tacrolimus-based maintenance immunosuppression, and rejections were not associated with severe BKPyV DNAemia.

In a subset analysis investigating KT recipients with available data on HLA-ABDR matching, HLA-ABDR matching had no impact on the risk of any BKPyV DNAemia. However, both the number of HLA-ABDR mismatches (HR per mismatch, 1.21; 95% CI 1.03–1.42; *p* = 0.024) and >3 HLA-ABDR mismatches (HR, 1.86; 95% CI 1.13–3.08; *p* = 0.015) were associated with an increased risk of severe BKPyV DNAemia. This association remained statistically significant in the adjusted models. Furthermore, when adjusting for HLA-ABDR mismatches, living donors were no longer associated with an increased risk of severe BKPyV DNAemia (HR, 2.79; 95% CI 0.95–8.14; *p* = 0.061).

### 3.4. Outcomes in KT Recipients with BKPyV DNAemia

Immunosuppressive treatment was reduced in 102 out of 129 kidney transplant (KT) recipients exhibiting BKPyV DNAemia. Mycophenolate mofetil (MMF) was reduced in 97 recipients, prednisolone in 36 recipients, calcineurin inhibitors in 28 recipients, and tacrolimus was switched to cyclosporine A (CyA) in one recipient. Additionally, azathioprine was reduced in three recipients, and everolimus was reduced in one recipient. In the remaining 27 KT recipients, no reduction in immunosuppressive agents was deemed necessary, as the planned tacrolimus reduction was considered sufficient.

In total, 24 (4%) of the KT recipients experienced graft function decline within the first two years post-KT, 10 (42%) of which also had any BKPyV DNAemia. In 14 (58%) KT recipients, graft function decline progressed to graft loss, six (43%) of which also had any BKPyV DNAemia. Time-dependent Cox proportional hazards models were used to investigate if any BKPyV DNAemia and severe BKPyV DNAemia were associated with an increased risk of graft function decline, graft loss, or death (Figure 3). In the unadjusted model, any BKPyV DNAemia was associated with an increased risk of graft function decline (HR, 2.48; 95% CI 1.10–5.57; *p* = 0.028), and severe BKPyV DNAemia was associated with an increased risk of graft loss (HR, 3.5; 95% CI 1.17–10.42; *p* = 0.025). In the model adjusted for recipient age, any BKPyV DNAemia was consistently associated with an increased risk of graft function decline (aHR, 2.26; 95% CI 1.00–5.12; *p* = 0.049), and severe BKPyV DNAemia was consistently associated with an increased risk of graft loss (aHR, 3.18; 95% CI 1.06–9.58; *p* = 0.039). In total, 16 (3%) recipients died, and neither any BKPyV DNAemia nor severe BKPyV DNAemia were associated with an increased risk of death.

In the sensitivity analysis, a single BKPyV DNA estimate ≥ 10,000 copies/mL was investigated as a risk factor for the defined adverse outcomes. In this analysis, we found a single PCR estimate ≥ 10,000 copies/mL to be associated with an increased risk for graft loss in both the unadjusted (HR, 3.48; 95% CI 1.22–10.21; *p* = 0.021) and the adjusted models (HR, 3.15; 95% CI 1.08–9.42; *p* = 0.038). A single BKPyV DNA estimate ≥ 10,000 copies/mL was not associated with an increased risk of graft function decline or death.

Graft biopsies were performed in 74 (57%) KT recipients with any BKPyV DNAemia, 14 (20%) of which had BKPyVAN. Graft biopsies were performed in 23 (96%) KT recipients with graft function decline and 14 (100%) KT recipients with graft loss within the first two years post-KT. BKPyVAN was found in three (13%) KT recipients with graft function decline and two (14%) KT recipients with graft loss.

## 4. Discussion

In this study, we investigated a large, prospective cohort of KT recipients that underwent routine screening for BKPyV, and we determined the cumulative incidence of BKPyV DNAemia; the risk factors for BKPyV DNAemia; and whether BKPyV DNAemia was associated with graft function decline, graft loss, and death. We found that one in five KT recipients developed BKPyV DNAemia during the first year post-KT, and that severe BKPyV DNAemia was frequent. Recipient age above 50 years at transplantation, male sex, living donors, and a higher degree of HLA-ABDR mismatches were associated with an increased risk of severe BKPyV DNAemia post-KT. Furthermore, BKPyV DNAemia increased the risk of graft function decline, and severe BKPyV DNAemia increased the risk of graft loss.

We found a cumulative incidence of 21% and 13% for BKPyV DNAemia and severe BKPyV DNAemia, respectively. These findings align with findings from other centers [9,12]. However, direct comparisons between studies are challenging due to variations in quantitative assays and the related cut-offs. Moreover, the reported incidences may be influenced by variations in protocols for immunosuppressive therapy [12], ureteral stent usage [3,11], and screening approaches [26].

We found that recipient age and male sex were associated with an increased risk of severe BKPyV DNAemia. Increased recipient age is an important risk factor for BKPyV DNAemia, which is supported by previous studies [10,12]. Likewise, male sex has been widely recognized as a significant risk factor for BKPyV DNAemia [10,12,27,28]. Furthermore, we found that both a higher number of HLA-ABDR mismatches and more than three mismatches increased the risk of severe BKPyV DNAemia. Moreover, we found that KT recipients with a higher degree of HLA-ABDR mismatches were more likely to progress from any BKPyV DNAemia to severe BKPyV DNAemia. This is in accordance with previous studies, which have found HLA-ABDR mismatches to be an important risk factor for BKPyV DNAemia [8,27]. In contrast to previous studies [28], we found an increased risk of severe BKPyV DNAemia in recipients receiving allografts from living donors. This could be explained by a higher degree of HLA-ABDR mismatches in living donor KT, necessitating more intensive immunosuppressive treatment [12,15]. We found no association between cold ischemia time and BKPyV DNAemia. This contrasts with a previous study that linked BKPyV DNAemia to BKPyVAN [29]. These results support the hypothesis that the risk factors for BKPyV DNAemia and BKPyVAN may differ. Rejection treatment and BKPyV DNAemia are known to be risk factors for one another [8,30]; however, we did not find an increased risk of BKPyV DNAemia after rejection treated with methylprednisolone, which is in accordance with another study using the same methodical design [27].

We found KT recipients with BKPyV DNAemia to be at an increased risk of graft function decline and recipients with severe BKPyV DNAemia to be at an increased risk of graft loss but not an increased risk of death within two years post-KT. In a previous study, it has been suggested that any BKPyV-related outcome post-KT, including rejection, infections, or graft failure, is accompanied by BKPyVAN [31]; however, this study did not include graft function decline as an outcome. A reduction in immunosuppression is advised when BKPyV DNA is persistently greater than 10,000 copies/mL due to a significant risk of BKPyVAN [18,24]. In line with our findings, previous studies have shown a greater occurrence of graft loss in kidney transplant recipients with high-level BKPyV DNAemia, defined as BKPyV DNA greater than 10,000 copies/mL [7]. However, it is worth noting that, even in cases with BKPyV DNA below 10,000 copies/mL, BKPyV DNAemia may progress to BKPyVAN [19]. Besides the viral load, the duration of BKPyV DNAemia is a possible risk factor for adverse outcomes in recipients with BKPyV DNAemia. One study found patients with persistent high viremia, defined as ≥3 months of BKPyV DNAemia with a peak of ≥10.000 copies/mL, to be at an increased risk of graft loss and decreased graft function compared to those with transient high viremia, defined as <3 months with peak BKPyV DNAemia ≥ 10.000 copies/mL, within a median of three years post-KT [4]. Another study found that persistent viremia, defined as BKPyV DNAemia > 400 copies/mL for ≥140 days, was not associated with an increased risk of allograft or patient survival within a median of three years post-KT [32]. To further investigate the impact of BKPyV DNAemia duration, we investigated the impact of two consecutive BKPyV DNA estimates ≥ 10.000 copies/mL. In agreement with previous studies, we found that the BKPyV DNA estimate of ≥10,000 copies/mL was associated with an increased risk of graft loss. Importantly, two consecutive BKPyV DNA estimates ≥ 10,000 copies/mL did not yield additional precision in predicting adverse outcomes.

The present study has notable strengths, including a large study population of KT recipients. The routine screening program ensured regular screening for BKPyV DNAemia, and the BKPyV PCR results were collected in the MiBa database, which encompasses all microbiological samples conducted in Denmark.

However, some limitations must be acknowledged. One limitation was the infrequent schedule for BK screening compared to some other studies, which could have resulted in the delayed detection of some cases of BKPyV DNAemia. While there was a general adherence to the screening program until graft loss, death, or end of follow-up, a small subset of kidney transplant (KT) recipients did not fully comply with the scheduled testing regimen. Furthermore, we did not assess the impact of immunosuppressive reduction in relation to BKPyV DNAemia, which may have increased the risk of graft function decline or graft loss due to rejections. An additional limitation in our study was the absence of routine graft biopsies in cases of BKPyV DNAemia, which limited the diagnosis of BKPyVAN. However, it is worth noting that patients experiencing both graft function decline and graft loss underwent thorough investigation, including graft biopsies.

## 5. Conclusions

In conclusion, we found a high incidence of BKPyV DNAemia during the first year following kidney transplantation, affecting approximately one in five recipients, with severe BKPyV DNAemia occurring in one out of eight recipients. Notably, older recipient age, male sex, having a living donor, and a higher degree of HLA-ABDR mismatches were risk factors for severe BKPyV DNAemia. BKPyV DNAemia was associated with an increased risk of graft function decline, while severe BKPyV DNAemia was associated with a higher risk of graft loss. These results emphasize the importance of early and regular monitoring for BKPYV infection, as well as the need for intensive clinical management, to prevent adverse outcomes in kidney transplant recipients.

Future studies should explore the potential impact on graft function associated with the reduction of immunosuppression in KT recipients with BKPyV DNAemia.

## Figures and Tables

**Figure 1 microorganisms-12-00065-f001:**
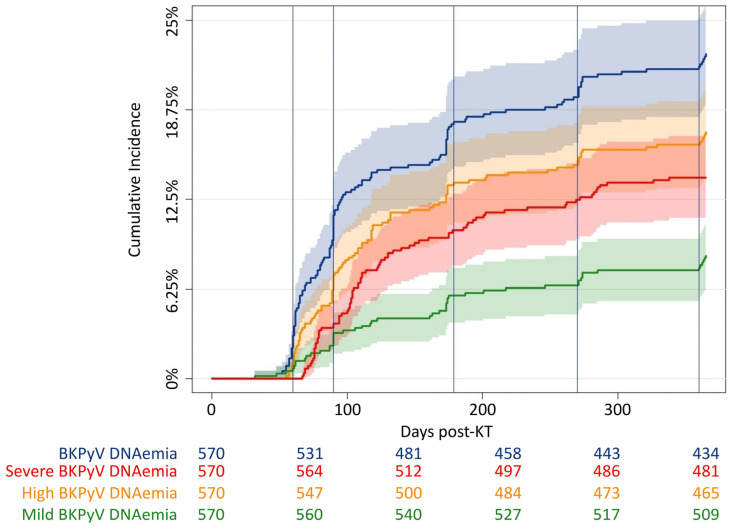
Cumulative incidence curves of BKPyV DNAemia. Cumulative incidence of any BKPyV DNAemia (blue), severe BKPyV DNAemia (red), high BKPyV DNAemia (yellow, a single BKPyV DNA estimate ≥ 10,000 copies/mL), and mild BKPyV DNAemia (green, a single BKPyV DNA estimate ≥ 10,000 copies/mL without progression to severe BKPyV DNAemia) within the first year post-KT. Vertical lines at scheduled screening days: at days 30, 60, 180, 270, and 360, respectively.

**Figure 2 microorganisms-12-00065-f002:**
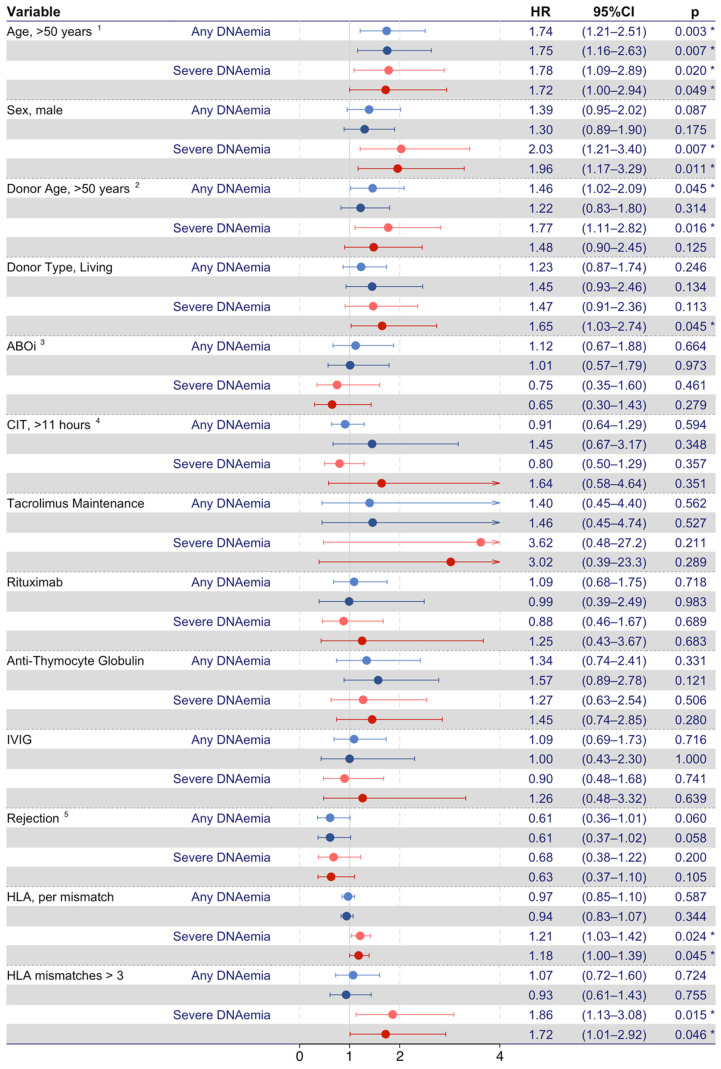
Time-dependent Cox proportional hazards models on risk factors for BKPyV DNAemia. Forest plots represent risk factors for any BKPyV DNAemia (light and dark red-colored forest plots mark unadjusted and adjusted values, respectively) and severe BKPyV DNAemia (light and dark blue-colored forest plots mark unadjusted and fully adjusted values, respectively). ^1^ Recipient age > 50 years at time of transplantation. ^2^ Donor age > 50 years at time of transplantation. ^3^ ABOi, ABO incompatible donor. ^4^ Cold ischemia time > 11 h, the time between the chilling and removal of the donor kidney and the time of restored blood supply in the recipient. ^5^ Rejection as a time- dependent covariate. The asterisk (*) represents *p* < 0.05.

**Figure 3 microorganisms-12-00065-f003:**
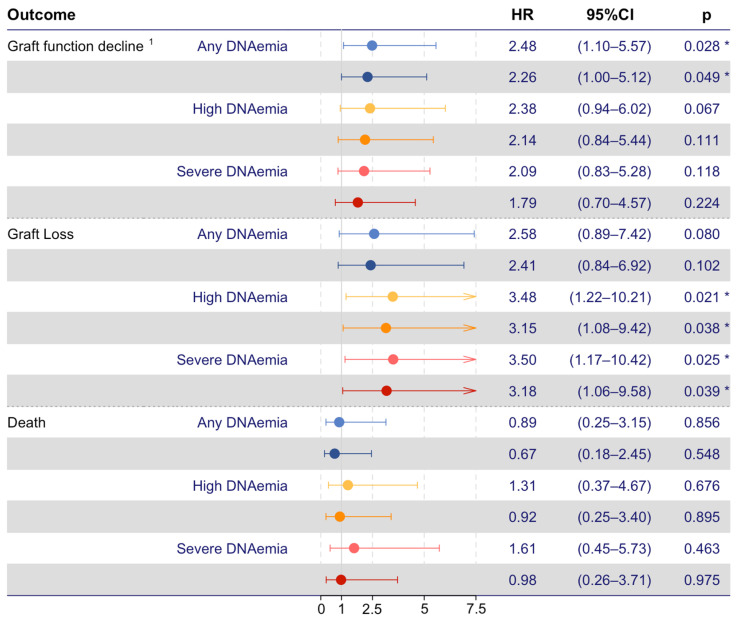
Time-dependent Cox proportional hazard models on recipient outcomes. BKPyV DNAemia and severe BKPyV DNAemia as risk factors for poor outcomes (red and blue, respectively. Dark-colored forest plots mark age-adjusted values). The sensitivity analyses, including high BKPyV DNAemia defined as a single BKPyV DNA measurement *≥* 10,000 copies/mL, are shown in yellow forest plots. ^1^ Graft function decline, defined as graft loss or 50% decline in eGFR from day 90 post-KT until the end of follow-up. The asterisk (*) represents *p* < 0.05.

**Table 1 microorganisms-12-00065-t001:** Cohort demographics of the 570 investigated recipients stratified by BKPyV DNAemia status.

KT Characteristics	All *n* = 570	No DNAemia *n* = 441	Any DNAemia *n* = 129	*p*	Mild DNAemia *n* = 49	Severe DNAemia *n* = 80	*p*
Age at KT, median (IQR)	50 (41–61)	49 (40–60)	54 (44–63)	0.008 *	51 (39–59)	56 (46–65)	0.007 *
Sex, male (%)	366 (64)	275 (62)	91 (71)	0.109	31 (63)	60 (75)	0.222
KT number, *n* (%)				1.000			0.982
1st transplant	497 (87)	385 (87)	112 (87)		42 (86)	70 (88)	
≥2nd transplant	72 (13)	56 (13)	17 (13)		7 (14)	10 (13)	
Disease leading to KT, *n* (%)				0.472			NA
Glomerulonephritis	149 (26)	119 (27)	30 (23)		11 (22)	19 (24)	
Cystic kidney diseases	98 (17)	80 (18)	18 (14)		6 (12)	12 (15)	
Diabetic nephropathy	48 (8)	37 (8)	11 (9)		5 (10)	6 (8)	
Vascular & Hypertension	25 (4)	18 (4)	7 (5)		<5 (10)	<5 (6)	
Systemic diseases	33 (6)	25 (6)	8 (6)		<5 (10)	<5 (6)	
Other ^1^	76 (13)	57 (13)	17 (13)		9 (18)	8 (10)	
Unknown	141 (25)	105 (24)	38 (29)		10 (20)	28 (35)	
Maintenance immunosuppression at discharge, *n* (%) ^2^				0.735			NA
Tacrolimus-MMF-Pred	549 (96)	423 (96)	126 (98)		47 (96)	79 (99)	
CyA-MMF-Pred	12 (2)	10 (2)	<5 (4)		<5 (10)	0 (0)	
mTOR-MMF-Pred	<5 (1)	<5 (1)	0 (0)		0 (0)	0 (0)	
Other combinations ^3^	<5 (1)	5 (1)	<5 (4)		<5 (10)	<5 (6)	
Induction therapy, *n* (%) ^4^							
Basiliximab	552 (94)	429 (95)	123 (95)	0.168	49 (100)	74 (92)	0.509
IVIG	96 (16)	73 (16)	23 (18)	0.902	<5 (10)	13 (16)	0.945
Rituximab	92 (16)	70 (16)	22 (17)	0.921	10 (20)	12 (15)	0.797
Anti-thymocyte Globulin	37 (6)	26 (6)	11 (9)	0.388	<5 (10)	8 (10)	0.660
Donor age, median (IQR)	54 (44–64)	54 (44–64)	57 (46–64)	0.132	56 (49–61)	58 (45–67)	0.491
Deceased donor, *n* (%)	323 (57)	255 (58)	68 (53)	0.353	32 (65)	36 (45)	0.629
ABOi, *n* (%) ^5^	65 (11)	49 (11)	16 (12)	0.804	8 (16)	8 (10.0)	0.434
HLA-ABDR mismatches ^6^							
Total, median (IQR)	3 (2–4)	3 (2–4)	3 (2–4)	0.629	2 (0–3)	3 (2–4)	0.009 *
>3 mismatches, *n* (%)	148 (27)	114 (27)	34 (27)	1.000	7 (15)	27 (34)	0.030 *
Cold ischemia time (hours), median (IQR)	11 (3–18)	11 (3–18)	10 (3–19)	0.749	6 (3–15)	11 (3–19)	0.348

^1^ Other diseases leading to KT included chronic interstitial nephritis, reflux nephropathy, pyelonephritis, hereditary and congenital urethral anomalies, Alport syndrome, etc. ^2^ Immunosuppressive treatment received at the time of discharge post-KT. Twenty recipients were censored before discharge. MMF, mycophenolate mofetil. Pred, prednisolone. CyA, cyclosporine A. mTOR, mammalian target of rapamycin. ^3^ Other immunosuppressive combinations included azathioprine and dual-drug therapy. ^4^ Induction therapy used in combination with methylprednisolone. ^5^ ABOi, ABO incompatible donor. ^6^ HLA-ABDR matching was only available in 558 of the KT recipients. The asterisk (*) represents *p* < 0.05.

**Table 2 microorganisms-12-00065-t002:** BK screening results for 570 KT recipients included in screening program.

Parameters	Results
BKPyV PCR performed in all recipients, *n*	3577
BKPyV PCR per recipient, median (IQR)	5 (4–7)
Total number of negative BKPyV PCR, *n* (%)	2722 (76)
Total number of positive BKPyV PCR, *n* (%)	855 (24)
BKPyV PCR performed in BKPyV-negative recipients, *n*	2143
BKPyV PCR per recipient, median (IQR)	5 (4–6)
BKPyV PCR performed in BKPyV-positive recipients, *n*	1493
BKPyV PCR per recipient, median (IQR)	11 (9–13)
Total number of negative BKPyV PCR, *n* (%)	638 (43)
Total number of positive BKPyV PCR, *n* (%)	855 (57)
Time of first BKPyV detection, median (IQR) ^1^	90 (63–145)
Time of last BKPyV detection, median (IQR) ^1^	261 (161–338)
Peak viral load, median (IQR) ^2^	1.7 × 10^5^ (2.6 × 10^4^–1.9 × 10^6^)

^1^ Time of first and last BKPyV-positive PCR in days post-KT. ^2^ Peak viral load, measured in copies/mL.

## Data Availability

The data from this study are accessible for review in person at our institution by request. Unfortunately, the data cannot be made publicly available in compliance with Danish data protection laws.

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
