# Peer review of "BKPyV DNAemia in Kidney Transplant Recipients Undergoing Regular Screening: A Single-Centre Cohort Study"

_microorganisms, 2023, doi:10.3390/microorganisms12010065_

Round 1
Reviewer 1 Report
Comments and Suggestions for Authors
BK virus remains a significant issue for clinicians- its monitoring and management are open to further improvement. A Single centre prospective study that is well performed and comprehensive provide some benefit for understanding care.
In the present study the authors describe associations between arbitrary levels of BK viraemia (BKV) and what they term severe BK viraemia and present data up to 2 years out for analysis with graft function decline and graft and patient outcomes. They confirm that BKV associates with worse graft function change and the more severe the viraemia the worse the association now including graft loss but not death. They note risk being associated with older recipient age and live donors. They don't provide data on management/treatment other than indicating an individualised dose reduction hierarchy of change but no data on how this was actually applied nor any response to such changes (unfortunately). I have some comments suggestions:
1) BKV was any level of plasma BK >1000 - is this on any single occasion or more than 1? Is that correct? Please clarify
2) Instead of analysing peak BK or an estimate of exposure (time > 10^4 for example) they choose the cut-off of 2 estimates > 10^4 on consecutive testing- can the authors justify this definition of severity or reference it? Is it in house or known to be evidence based? Did you consider exploring a level as a cut-off to associate with graft decline or loss? Perhaps this should be discussed under methods/stats?
3) What method (commerical/in house) for estimation of viral load was used and is it serum or plasma and was it unchanged during the time of the study
4) Was HLA matching available? I would have assumed so but its not provided- in order to discuss that "living donors might have higher mismatches" as a reason for higher risk of more severe BK (which I am less convinced is correct as 75% of live donors would be expected to be HLA matched or 1 haplotype matched?) this data should be provided please or an explanation of why it was unavailable- add to limitation please.
5) timing of BKV and rejection- we aren't provided with any idea of whether BK precedes or occurs after rejection- it may be a two way relationship so the statement in the discussion p 8 lines 247/248 regarding 'we didn't find an increased risk of BKV after rejection treated with MP" might need to be corrected or the data regarding this timing provided to allow it to be substantiated?
6) The table showing the 129 with BKV and the 80 with severe BKV is presented such that patients are reported twice- the 80 severe BKV are also presented within the 129 BKV- is that correct? This means showing the data for the other features overlaps and isn't that informative (you don't perform any statistical comparison) - is it better to show the 49 with BKV and the 80 with severe BKV separately in order to see if there are differences which are otherwise concealed including the peak BK viral load and range? You can then still combine them when required - this might also effect Figure 1. The reason being that is there something different about those who dont progress to severe BKV that we are missing- have you examined this? Please consider
7) Graft loss - it seems disappointing that you cannot provide evidence for biopsy proven BKVAN or graft loss to include here- I accept that the association is shown but its still a significant limitation- because graft loss could be BKVAN or rejection associated with immunosuppression rejection (either is graft failure but better precision would make this more impressive study).
8) Is there any data to add regarding whether changes to immunosuppression as outlined as part of treatment approach were actually made? This too might be acknowledged as a limitation because again we cant be certain how this impacts the outcomes of graft decline/loss?
9) The limitations are I think a little more than as written (see above comments) the strengths are accepted .
Small point: Table 1 the numbers/distribution for maintenance suppression at discharge n (5) for the BKV and severe BKV dont seem to add correctly- please check e.g 7/80 severe BKV shown as 90% you mean 99%?
Reviewer 2 Report
Comments and Suggestions for Authors
I read with attention the article from Rasmussen et al. that describes BK Polyomavirus-related outcomes in a Danish large cohort. The article is well written and the results are wisely presented, overall. However, I noticed flaws in the analysis that can be corrected by minor and major revisions. No experimental analysis is required. I suggest an important edition of the manuscript and a new processing of the data to make the conclusions fully concordant with the results.
According to latest recommendations, the terms “BKV” “BKVN” and “viremia” should be replaced by “BKPyV”, “BKPyVAN” and “DNAemia”, respectively
Definition of BKV viremia is not consensual since it includes only BKV DNA > 1000 copies/mL and not below. Authors must justify their choice and they should provide a formal analysis of BKV DNaemia as commonly defined.
Since the included patients have not the exact same follow-up period (some patients may have been followed for a shorter period until death or graft loss), the study may be biased, and conclusions could be affected by this methodological flaw. Data should be analyzed again but with patients with the same follow-up criteria only. It may reduce the size of the cohort though.
Ine 109: how do the authors define “significant decrease in viral load” ?
For some results, the authors reported the precise p value. For others, they reported “<0.05” (example: line 186). Please precise the p value for the whole statistical analyses.
Results (3.2 and table 2) : the fact that some (how many) patients had only 1 BKV PCR is contrary to the methods (“This screening program included BKV PCR tests for in plasma conducted on day 60, 90, 180, 270 and 360 post-KT [21]. Following detection of BK viremia, additional BKV PCR tests were conducted every two weeks until a significant decrease in viral load was observed.”). The minimum number of BKV PCRs should be 5.
Lines 160-161: pay attention to values editing.
Lines 241-242: a multivariate analysis would adjust this bias ?
Line 261: since the authors closely monitored DNaemia, an analysis of DNAemia duration would be very interesting.
Round 2
Reviewer 2 Report
Comments and Suggestions for Authors
I would like to thank the authors of the manuscript for their corrections and their responses to my comments. The revised version of the manuscript is greatly improved and the limitations of the study are sufficiently discussed. The article is rich in data that will probably be of great help for future studies on this topic.